# Genomic insights into methicillin-resistant *Staphylococcus pseudintermedius* isolates from dogs and humans of the same sequence types reveals diversity in prophages and pathogenicity islands

**Nathita Phumthanakorn**[1,2,3]**, Sybille Schwendener**[2]**, Valentina Donà**[2]**, Pattrarat Chanchaithong**[1,4]**, Vincent Perreten**[2�]*****, Nuvee Prapasarakul**[1,4☸]*****

**1** Department of Veterinary Microbiology, Faculty of Veterinary Science, Chulalongkorn University, Bangkok, Thailand, **2** Institute of Veterinary Bacteriology, Vetsuisse Faculty, University of Bern, Bern, Switzerland, **3** Department of Pre-clinic and Applied Animal Science, Faculty of Veterinary Science, Mahidol University, Nakhon Pathom, Thailand, **4** Diagnosis and Monitoring of Animal Pathogens Research Unit, Chulalongkorn University, Bangkok, Thailand

☸ These authors contributed equally to this work.
* vincent.perreten@vetsuisse.unibe.ch (VP); nuvee.p@chula.ac.th (NP)

**Data Availability Statement:** The whole genome sequences of the S. pseudintermedius strains are

## Abstract

Methicillin-resistant *Staphylococcus pseudintermedius* (MRSP) is an important opportunistic pathogenic bacterium of dogs that also occasionally colonize and infect humans. However, whether MRSP can adapt to human hosts is not clear and whole genome sequences of MRSP from humans are still limited. Genomic comparative analyses of 3 couples of isolates from dogs (n = 3) and humans (n = 3) belonging to ST45, ST112, and ST181, the dominant clones in Thailand were conducted to determine the degree of similarities between human and animal MRSP of a same ST. Among eight prophages, three prophages associated with the leucocidins genes (*lukF/S-I*), φVB88-Pro1, φVB16-Pro1 and φAP20-Pro1, were distributed in the human MRSPs, while their remnants, φAH18-Pro1, were located in the dog MRSPs. A novel composite pathogenicity island, named SpPI-181, containing two integrase genes was identified in the ST181 isolates. The distribution of the integrase genes of the eight prophages and SpPI-181 was also analysed by PCR in 77 additional MRSP isolates belonging to different STs. The PCR screen revealed diversity in prophage carriage, especially in ST45 isolates. Prophage φAK9-Pro1 was only observed in ST112 isolates from dogs and SpPI-181 was found associated with ST181 clonal lineage. Among the 3 couple of isolates, ST45 strains showed the highest number of single nucleotide polymorphisms (SNP) in their core genomes (3,612 SNPs). The genomic diversity of ST45 isolates suggested a high level of adaptation that may lead to different host colonization of successful clones. This finding provided data on the genomic differences of MRSP associated with colonization and adaption to different hosts.

available from the Genbank database [accession numbers CP030374 (AH18-ST45), CP030715 (VB88-ST45), CP031603 (AK9-ST112), CP031605 (VB16-ST112), CP031604 (AI14-ST181), and CP031561 (AP20-ST181)].

**Funding:** This project was financed by the Overseas Research Experience Scholarship for graduate school to Nathita Phumthanakorn, Ratchadapisek Sompoch Endowment Fund (CU-GR_61_012_31_002) Chulalongkorn University, and from internal funds of the Institute of Veterinary Bacteriology, University of Bern, Switzerland. The funders had no role in study design, data collection and analysis, decision to publish, or preparation of the manuscript.

**Competing interests:** The authors have declared that no competing interests exist.

## Introduction

Methicillin-resistant *Staphylococcus pseudintermedius* (MRSP) is an important bacterial pathogen of dogs. The treatment of MRSP infections e.g., canine pyoderma and otitis is often a challenge due to the multidrug resistance traits of the strains which limit therapeutic options [1]. MRSP from dogs can also be transmitted to humans, especially to those who are in close contact with dogs like dog's owners and veterinarians [2, 3]. In dog-owning household members, *S. pseudintermedius* can persist up to one year [4]. *S. pseudintermedius* also occasionally cause infections in humans such as skin and soft tissue infections, dog bite wound infection, and rhinosinusitis [5–7]. By multilocus sequence analysis, MRSP ST45, ST112 and ST181 are predominant in dogs and can be isolated from people associated with dogs in Thailand that are different from major clones in Europe (MRSP ST71) and North America (MRSP ST68) [3, 4].

Previously, human-specific adaptation has been observed for livestock-associated *S. aureus* ST398, and involved changes in the surface protein genes and carriage of specific prophages [8, 9]. However, little is known about the genetic properties, which differentiate *S. pseudintermedius* clones from the same ST but isolated from different sources. Advances in next generation sequencing (NGS) permit nowadays to obtain accurate complete genome sequences identifying both accessory genetic elements and single nucleotide polymorphisms (SNPs) [10]. The combination of long and short-read NGS technologies, like Oxford Nanopore Technologies (ONT, Oxford Nanopore Technologies, Oxford, UK) with Illumina® (Illumina, San Diego, CA), generates accurate complete genomes of bacteria, which results from the genome scaffold obtained with the long reads of ONT and corrected with the accurate sequence of Illumina [11, 12]. Additionally, long read sequencing platform provides an important role for the study of structural variation (SV), including insertions, deletions, duplications, inversions, and large-scale structural rearrangements in short sequence variant [11].

In this study, we used these sequencing approaches to compare whole genome sequences of MRSP isolates from a couple of dogs and humans belonging to the same ST. The comparative genome analysis could demonstrate the genetic diversity related to host origin and presumptive catalogue gene set in MRSPs that might be associated with the host adaptation.

## Materials and methods

### Bacterial isolates

Six representative MRSP isolates previously collected and characterized from dogs [n = 3: AH18 (ST45), AK9 (ST112), and AI14 (ST181)] and humans [n = 3: VB88 (ST45), VB16 (ST112), and AP20 (ST181)] were included for whole genome sequencing and analysis [3]. Canine MRSP strain AH18 (ST45) was isolated from nasal mucosa, and MRSP strain AK9 (ST112) and AI14 (ST181) were isolated from groins. Human MRSP strain VB16 (ST112) and VB88 (ST45) were recovered nares from small animal veterinarians, and MRSP AP20 (ST181) was isolated from nares of a dog owner. The collection was conducted at the Small Animal Teaching Hospital, Faculty of Veterinary Science, Chulalongkorn University in 2010–2012. By 85% similarity cut-off of *Sma*I-PFGE macrorestriction analysis, canine and human MRSP ST112 (strain AK9 and VB16) were clustered into the same group but showed the different PFGE patterns. Two MRSP ST181 (strain AI14 and AP20) were in distinct groups. Canine and human MRSP ST45 (strain AH18 and VB88) were in different groups by *Cfr*9I-PFGE. The bacteria were stored in tryptic soy broth with 25% glycerol at -80˚C and recovered on tryptic soy agar with 5% sheep blood.

Seventy-seven additional MRSP isolates from dogs (n = 61) and humans (n = 16) were included for integrase gene detection [3]. They were collected during 2010–2012 and belonged

to 17 sequence types (ST) present in Thailand. Isolates were kept in 25% glycerol stocks at -80˚C. The strains were obtained from a previous study [3], where the sampling of dogs were performed after the informed consent were completed and signed by the dog' owners. The ethical statement of dogs and humans were approved as previously described [3].

## Genome sequencing, assembly and annotation

Genomic DNA was extracted using DNeasy Blood & Tissue Kit (Qiagen, Hilden, Germany) following the manufacturer's protocol. Whole-genome sequencing was performed on MinION (Oxford Nanopore Technologies (ONT), Oxford, UK) and Illumina (Illumina Inc, San Diego, US) platforms [12]. Sequencing libraries for MinION were prepared from mechanically fragmented DNA (g-TUBE, Covaris) using the ONT 1D ligation sequencing kit (SQK-LSK108) with the native barcoding expansion kit (EXP-NBD103) (Oxford Nanopore). MinION sequencing was done on a R9.4 SpotON flow cell with a MinION MK1b device. The generated fast5 ONT reads were base-called and demultiplexed using ONT Albacore software v2.0.1 and assembled *de novo* with Canu v1.3 [13]. For error correction, paired-end reads from Illumina MiSeq (isolates belonging to ST45; Omics Sciences and Bioinformatics Center, Chulalongkorn University, Thailand) and HiSeq (isolates belonging to ST112 and ST181; Eurofins, Konstanz, Germany) were mapped against long read assembly with Geneious v10.1.3 [14]. Unmapped regions were validated by PCR and Sanger sequencing. Genome annotation was performed with Prokka v1.12 software [15].

## Genomic comparison, pan and core genome analysis

For pan genome analysis, Roary v3.11.2 was used to identify core and accessory genes [16]. The average nucleotide identity (ANI) was analyzed in JSpeciesWS using ANI based on BLAST+ (ANIb) to get the degree of genomic similarity [17]. *S. pseudintermedius* strain NA45 (ST84), which showed the highest nucleotide similarity (99% with 94% coverage) to our strains by BLASTn, was used as a reference strain (Accession no. CP016072). The identification of SNPs and phylogenetic trees of core genome was performed using the CSI Phylogeny 1.4 server with default parameters [18]. The BLAST Ring Image Generator (BRIG) v. 0.95 was used to compare and visualize whole-genome sequences of strains [19].

Mobile genetic elements (MGEs) and antimicrobial resistance genes were characterized using the following online available bioinformatics tools. Prophages were predicted using the PHASTER web server [20, 21]. Pathogenicity Islands (PI) were searched for using the Island-Viewer 4 web server [22]. Transposons and insertion sequences (ISs) were searched from sequence annotation by Prokka v1.12 [15] and Blast searched in ISfinder database (https://www-is.biotoul.fr/index.php), respectively. Plasmids and bacterial resistome were predicted using PlasmidFinder 1.3 [23] and ResFinder 3.0 [24], respectively, available at the Center for Genomic Epidemiology (CGE) (https://cge.cbs.dtu.dk). The structures of MGEs, including prophages, plasmid, and PI predicted from web-based tools were manually inspected and edited. Therefore, predicted *att* sites were manually corrected by alignment of sequences up- and downstream of the prophage and PI region using Clustal W in Geneious v.10.1.3 [14, 25]. The size of prophages and PIs were referred to the sequence situated between identified attachment sequences (*att*) and/or the region associated with prophage and PI structure. Pairwise and multiple alignment were used to compare nucleotide and amino acid sequences of integrases, cell wall-associated (CWA) proteins or *S. pseudintermedius* surface protein genes (*sps*) protein by using Clustal W in Geneious v.10.1.3 [14]. Partial deletion of CWA protein genes (found from sequence alignment) were confirmed by PCR and Sanger sequencing. Primers used in this study are listed in S1 Table.

### Prophage- and PI-associated integrase gene detection in additional MRSP

The presence of integrase genes of prophages- and PI-associated gene was determined for a previously collected isolates (n = 77) [3]. The genomic DNA was extracted as previously described [27]. The detection was performed by PCR that was developed in this study (S1 Table).

### Accession number

The whole genome sequences of the *S. pseudintermedius* strains were deposited in Genbank under accession numbers CP030374 (AH18-ST45), CP030715 (VB88-ST45), CP031603 (AK9-ST112), CP031605 (VB16-ST112), CP031604 (AI14-ST181), and CP031561 (AP20-ST181).

## Results

### Genomic features of six representative MRSP

A total of six MRSP strains belonging to ST45, ST112, and ST181, three isolated from dogs and three counterparts isolated from humans, were used for the genomic analysis. The number of sequencing reads generated by MinION for MRSP strains AH18-ST45, VB88-ST45, AK9-ST112, VB16-ST112, AI14-ST181, and AP20-ST181 were 293,721, 595,311, 633,571, 268,283, 195,270, and 173,403 reads, respectively. The coverage of all strains was approximately 300- to 1000-fold and *de novo* assembly generated closed circular chromosomes for all strains. The genome sizes of the isolates were between 2.6–2.8 Mbp for all strains. Their general genomic features are presented in Table 1.

### Genomic comparison

From ANIb in JSpecies, the isolates of the same STs; namely the paired AH18-ST45 and VB88-ST45, AK9-ST112 and VB16-ST112, and AI14-ST181 and AP20-ST181, had 99.6%, 99.8%, and 99.8% genomic similarity between each other, consequently (S1 Fig). Genomic comparison between the paired dog and human MRSP strains of each ST revealed that the main differences were in the prophage integration regions (Fig 1). The other genetic differences of ST45 strains were found in the DpnII restriction-modification system (*dnpA* and *dpnM*), transporter system of TctABC transporter (*tctA*, *tctB*, and *tctC*), surface proteins Spa (*spsP* and *spsQ*), cell wall synthesis (*tagB*), and the other metabolic genes.

In addition to prophage sequences, the ST112 isolates differed by the presence of a 5.6 kbp putative restriction endonuclease of the LlaJI family (*llaJI*), which was detected in strain VB16-ST112 but absent in strain AK9-ST112. This fragment was composed of four genes encoding two putative restriction endonucleases, and two DNA-cytosine methyltransferases (*dcm*).

In addition to the different prophages, the human AP20-ST181 strain carried an element integrated into the iron ABC transporter substrate-binding protein (*fbpA*). This insert was associated with an IS*30* family transposase gene, had a length of 13.1 kbp and shared 97% nucleotide similarity (99% coverage, BLASTn) to *S. schleiferi* strain 2317–03 (accession no. CP010309). This transposon element carried genes encoding for secretory antigen SsaA-like protein (*ssaA*), membrane protein, cell division FtsK/SpoIIIE protein (*ftsk*/*spoIIIE*), conjugal transfer protein (*tra*), transcriptional regulator, and seven hypothetical proteins, respectively.

### Pan-genome analysis, SNP, and unique gene identification

The pairs of MRSP isolates belonging to the same ST shared similar staphylococcal cassette chromosome *mec* (SCC*mec*) and MGEs carrying antimicrobial resistance genes (Table 1). The

**Table 1. General features, mobile genetic elements (MGEs), antimicrobial resistance associated with MGEs, CWA protein genes, and CRISPR of six representative MRSP isolates.**

| | AH18-ST45 (dog) | VB88-ST45 (human) | AK9-ST112 (dog) | VB16-ST112 (human) | AI14-ST181 (dog) | AP20-ST181 (human) |
|---|---|---|---|---|---|---|
| | | | Strain-ST (source) | | | |
| **Features** | | | | | | |
| Genome size (bp) | 2,623,199 | 2,673,267 | 2,789,057 | 2,791,609 | 2,719,425 | 2,734,754 |
| GC content (%) | 37.5 | 37.5 | 37.3 | 37.5 | 37.5 | 37.5 |
| CDS | 2,427 | 2,513 | 2,610 | 2,655 | 2,533 | 2,571 |
| tRNA | 59 | 59 | 59 | 59 | 59 | 59 |
| tmRNA | 1 | 1 | 1 | 1 | 1 | 1 |
| miscellaneous RNA (miscRNA) | 54 | 58 | 65 | 67 | 67 | 67 |
| **MGEs** | | | | | | |
| Prophage | Remnant of φAH18-Pro1 | φ VB88-Pro1 / φ VB88-Pro2 | φ AK9-Pro1 | φ VB16-Pro1 / φ VB16-Pro2 / φ VB16-Pro3 | φ AI14-Pro1 | φ AP20-Pro1 |
| Pathogenicity islands | - | - | - | - | SpPI-181 | SpPI-181 |
| **Antimicrobial resistance genes and location** | | | | | | |
| SCC*mec* | ψSCC*mec*$_{57395}$ | ψSCC*mec*$_{57395}$ | SCC*mec*$_{AI16}$-SCC*czr*$_{AI16}$-CI | SCC*mec*$_{AI16}$-SCC*czr*$_{AI16}$-CI | SCC*mec*V(T) | SCC*mec*V(T) |
| Transposon (Tn) | | | | | | |
| • Tn*5405*-like | *aadE-sat4-aphA-3* and *erm*(B) | *aadE-sat4-aphA-3* and *erm*(B) | *aadE-sat4-aphA-3*, *erm*(B), and *dfrG* | *aadE-sat4-aphA-3*, *erm*(B), and *dfrG* | *aadE-sat4-aphA-3*, *erm*(B), and *dfrG* | *aadE-sat4-aphA-3*, *erm*(B), and *dfrG* |
| • Tn*552* | *blaZ, blaR1*, and *blaI-1* | *blaZ, blaR1*, and *blaI-1* | *blaZ, blaR1*, and *blaI-1* | *blaZ, blaR1*, and *blaI-1* | *blaZ, blaR1*, and *blaI-1* | *blaZ, blaR1*, and *blaI-1* |
| • Tn*916* | *tet*(M) | *tet*(M) | *tet*(M) | *tet*(M) | *tet*(M) | *tet*(M) |
| Integrated plasmid | *cat* | *cat* | - | - | - | - |
| Insertion sequence (IS) | | | | | | |
| • IS*256* | *aph(2")-Ia* | *aph(2")-Ia* | *aph(2")-Ia* | *aph(2")-Ia* | *aph(2")-Ia* | *aph(2")-Ia* |
| **CWA protein genes** | | | | | | |
| • *spsA-spsR* (18 genes) | *spsA-spsE, spsG-spsR* (17) | *spsA-spsE, spsG-spsO, spsR* (15) | *spsA-spsE, spsG-spsN, spsP-spsR* (16) | *spsA-spsE, spsG-spsN, spsP-spsR* (16) | *spsA-spsE, spsG-spsN, spsR* (14) | *spsA-spsE, spsG-spsN, spsR* (14) |
| **CRISPR subtype** | - | - | Type IIC | Type IIC | Type IIIA | Type IIIA |

AH18-ST45 and VB88-ST45 pair harbored ψSCC*mec*$_{57395}$, containing a class C1 *mec* gene and no *ccr* genes [26]. The AK9-ST112 and VB16-ST112 pair harbored SCC*mec*$_{AI16}$-SCC*czr*$_{AI16}$-CI, which consisted of a SCC*mec* with class A *mec* complex and *ccrA1B3* genes followed by a second SCC with *ccrA1B6*, and restriction modification and heavy metal resistance genes [27]. Finally, the AI14-ST181 and AP20-ST181 pair contained a SCC*mec* that was similar to SCC*mec* V(T) in the MRSP 23929 isolate from a dog in Ireland and SCC*mec* V of MRSP 063228 [28] and also possessed a class C2 *mec* gene complex, two *ccrC* genes (2 *ccrC8*) and a clustered regularly interspaced short palindromic repeats (CRISPR) and CRISPR-associated protein (Cas) gene complex.

Transposons and IS associated with antimicrobial resistance genes including Tn*5405*-like, Tn*916*, Tn*552*, and IS*256* families were observed in all investigated genomes (Table 1). Different gene structures of transposons and ISs were found in Tn*5405*-like elements (Table 1). Moreover, the transposase gene of the Tn*5405*-like element of isolates belonging to ST45 was classified as an IS*30*-like element IS*1216* family transposase (100% identity, BLASTx), while those of the ST112 and ST181 isolates were classified as an IS*1182* family transposase (98.2%

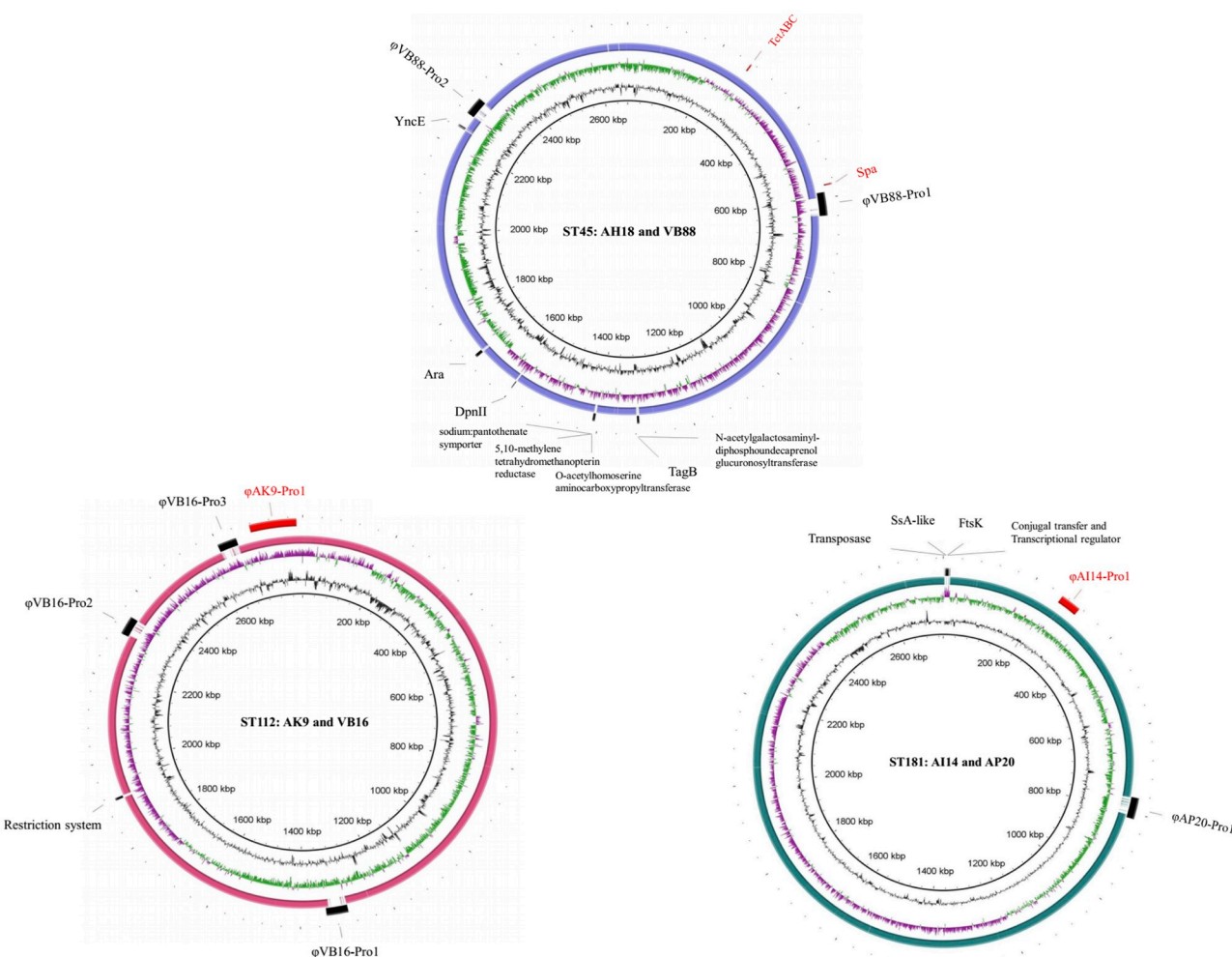

**Fig 1. Genomic comparison of three representative pairs of methicillin-resistant *S. pseudintermedius* (MRSP) strains isolated from dogs and humans belonging to ST45, ST112, and ST181.** Human strains were used as the reference. Circular diagram from inner to outer rings represent the % GC, GC skew (green-/ purple+), homology, location and difference in genes and prophages in human (black label) and dog (red label) isolates.

identity, BLASTx). Tn*5405*-like elements of isolates belonging to ST45 also possessed *aadE-sat4-aphA-3*, *erm*(B) but lacked *dfrG* in this element. Tn*5405*-like elements carrying *aadE-sat4-aphA-3*, *erm*(B) and *dfrG* coding for aminoglycosides, streptothricin, erythromycin, and trimethoprim resistances were found in isolates belonging to ST112 and ST181. Their structures were similar to the previously identified Tn*5405*-like elements of MRSP strain 69687 (ST71) and 23929 (ST68) [29]. An integrated plasmid carrying a chloramphenicol resistant gene (*cat*) was identified in the genome of the ST45 isolates. The *cat* and replication (*rep*) genes of the plasmid had 97.5% and 90.2% nucleotide similarity to pC221 (accession no. X02529.1), respectively, while the plasmid recombination enzyme (*pre*) was different from pC221 and had the highest similarity (100% nucleotide identity and coverage) to the *pre* type 2 gene in *S. aureus* strain GD1696 (accession no. CP040233).

The Restriction-Modification (RM) system and CRISPR were identified in the six genomes. The isolates of ST45 had a unique DNA methylase gene (*hsdM*) of Type I restriction system downstream of the ψSCC*mec* element and a homolog of the SsoII gene (type II endonucleases). The isolates of ST112 had a unique complete Type I restriction system containing *hsdR*, *hsdM*,

and *hsdS* in the SCC*czr*~AI16~ element and one copy of *hsdM* and *hsdS* in a different location in their genome. The isolates in ST181 contained two *hsdR* genes of the Type I RM system at different locations in the genome, and a gene encoding Type II restriction enzyme NgoFVII (endonuclease NgoFVII). Type II restriction enzyme Sau3AI (*sau3ai*) were identified in all the genomes except for VB88-ST45, while the restriction enzyme BgcI of the type II RM system were found in the isolates belonging to ST45. A Type II restriction enzyme homologous to MboI was identified in VB88-ST45 and the ST112 isolates, but not in AH18-ST45 and the ST181 isolates.

The CRISPR classification was performed by comparing the structure to that from a previous study in *S. pseudintermedius* [30]. CRISPR in isolates belonging to ST112 were type IIC, containing Cas1, Cas2, and Cas9 with 18 direct repeats (DRs). The CRISPR in the ST181 isolates were type IIIA containing nine *cas* genes for Cas1-2, Csm2-6, Cas6, and Cas10 with 16 DRs. However, no CRISPR system was identified in the ST45 isolates.

The phylogenetic relationship of the six MRSP isolates showed genetic relationship of the same ST strains, and the number of SNPs is shown in S2 and S3 Figs. A total of 13,037 SNPs from 2,327,265 positions were observed in all the analyzed core genomes. Pairs of isolates belonging to the same ST differed by 54 SNPs for ST181, 1,207 SNPs for ST112, and 3,612 SNPs for ST45 (S3 Fig). A higher diversity in the ST45 isolates was observed from the phylogenetic tree and agreed with the genomic similarity calculation from the JSpecies analysis (S1 Fig).

In addition, eight unique genes were predicted by Roary in the accessory genomes of the human MRSP strains only, which were an integrase, amidase, phage protein, and five hypothetical proteins that were located on different prophages (S2 Table). Moreover, *S. pseudintermedius* surface protein J (*spsJ*) genes were distinct in the human strains, where the *spsJ* genes of VB88-ST45, VB16-ST112, and AP20-ST181 shared 33%, 32%, and 28% amino acid identity, respectively, to SpsJ of *S. pseudintermedius* ED99 (accession no. CP002478.1). On the other hand, the region containing the *spsJ* sequence was annotated as a part of the *sasA* gene in the MRSPs from dog strains in the same ST (Fig 2). This difference resulted from a 4-bp deletion in the sequences of VB88-ST45 and VB16-ST112, and a 1-bp deletion in AP20-ST181 that caused a frameshift leading to a premature stop codon and alternative start codon for the *sasA* gene (Fig 2).

## Diversity of the putative CWA genes in ST45

The sets of CWA genes or *S. pseudintermedius* surface protein genes (*sps*) were analyzed in the isolates from human and dog sources. The *spsJ* gene has already been identified to be unique in human isolates (S2 Table). The other CWA protein genes or *sps* were determined based on data previously described in the genome of *S. pseudintermedius* ED99 [31]. The types, numbers, and sequence length of *sps* were conserved in each lineage, except for ST45 (Table 1). AH18-ST45 harbored two additional genes (*spsP* and *spsQ*) that were absent in VB88-ST45 and a different *spsO* genes. The *spsO* gene was 78 nucleotides longer in AH18-ST45 than in VB88-ST45. The different nucleotide sequences were confirmed by PCR and sequencing (S1 Table). Their full amino acid sequences were aligned to the reference strain ED99 and found to contain a repeat region of 26-aa (78-bp) (S4 Fig). The SpsO of AH18-ST45 contained four repeat regions, similar to ED99, while VB88-ST45 had three repeat regions according to this deletion.

## Diversity of prophages in MRSPs from dog and human

Altogether, there were eight prophages that ranged from 40–52 kbp in size with a 34.1–37.1% GC content, except for φAK9-Pro1 that was larger (~116 kbp) with a lower GC content

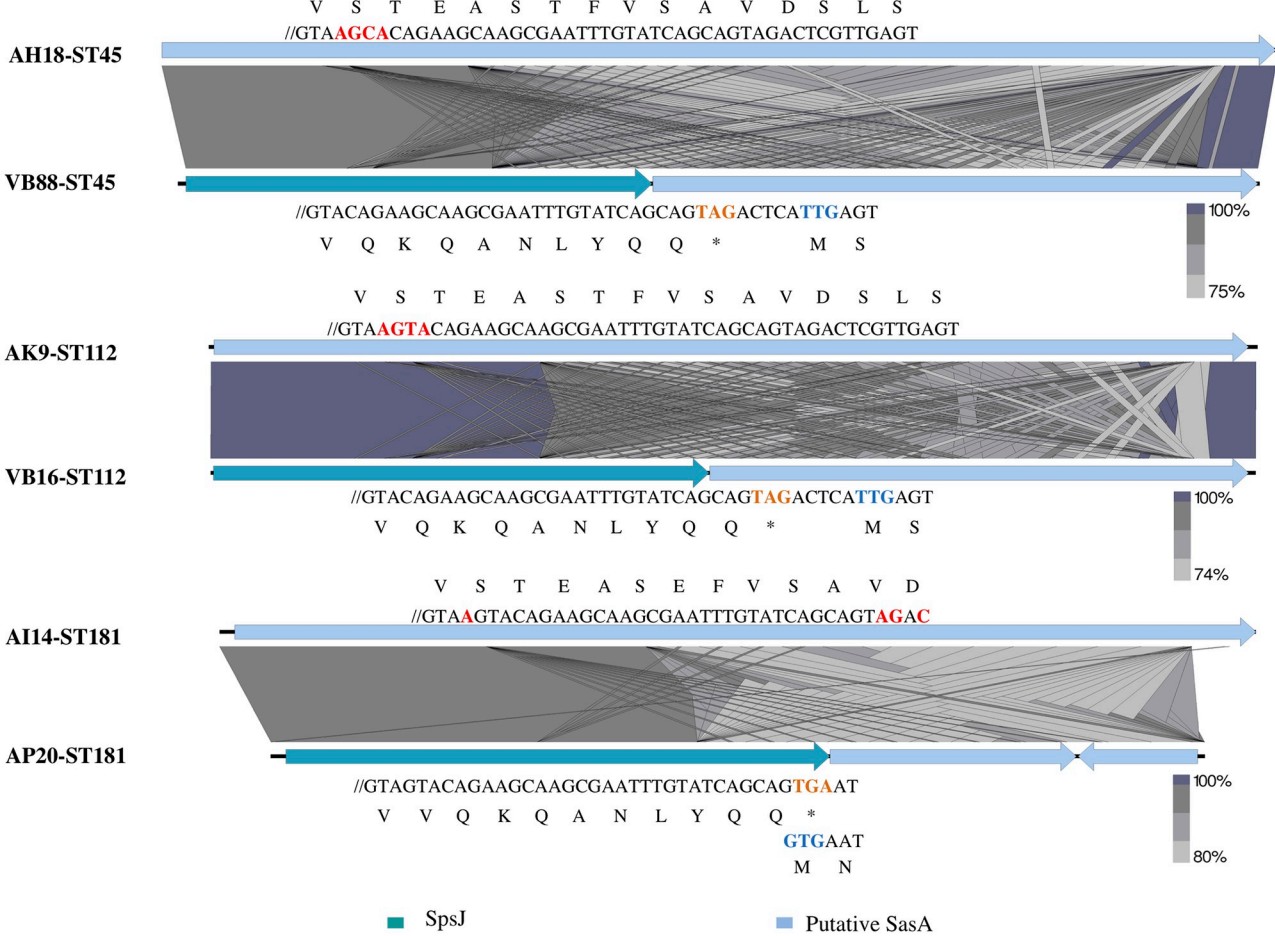

**Fig 2. Illustration of CWA protein genes diversity found in human isolates.** Comparison of *spsJ* and *sasA* gene in isolates from dogs and humans showing % nucleotide similarity, start codon (blue), premature stop codon (orange), and different nucleotide and amino acid sequence (red). Figure was generated using Easyfig 2.1.

(32.3%) (S3 Table). Interestingly, the three phages from the human isolates (φVB88-Pro1, φVB16-Pro1 and φAP20-Pro1) were integrated at the same chromosomal site, between the tRNA cluster gene and a phosphoglycerate kinase gene, and harbored identical integrases and *S. intermedius* leucocidins genes (LukI: *lukF/S-I*) (Fig 3). Overall, the prophages shared only 51.2% nucleotide similarity to each other and displayed different and shuffled modules, such as for replication and packaging. Phage φVB88-Pro1 was around 3 kbp larger in size than φVB16-Pro1 and φAP20-Pro1 and contained an insert associated with an *int* in the lysis modules. This integrase shared 100% amino acid identity to that of the phage remnant of AH18-ST45. A BLASTp search for phages similar to φVB88-Pro1, φVB16-Pro1 and φAP20-Pro1, identified the bacteriophage SpT99F3 of *S. pseudintermedius* ED99 (accession no. KX827371.1), which contained a phage integrase protein (*int*) with 70% amino acid similarity. Other components of SpT99F3 did not show any significant relatedness to the region of the three phages in the study and the *lukF/S-I* genes were absent. However, *lukF/S-I* was also present in all the dog MRSP isolates and was associated with a phage remnant in strain AH18 (remnant φAH18-Pro1) and phage-associated genes in strains AK9 and AI14 (Fig 3). The chromosomal location of *lukF/S-I* was also between the tRNA cluster and phosphoglycerate kinase encoding genes.

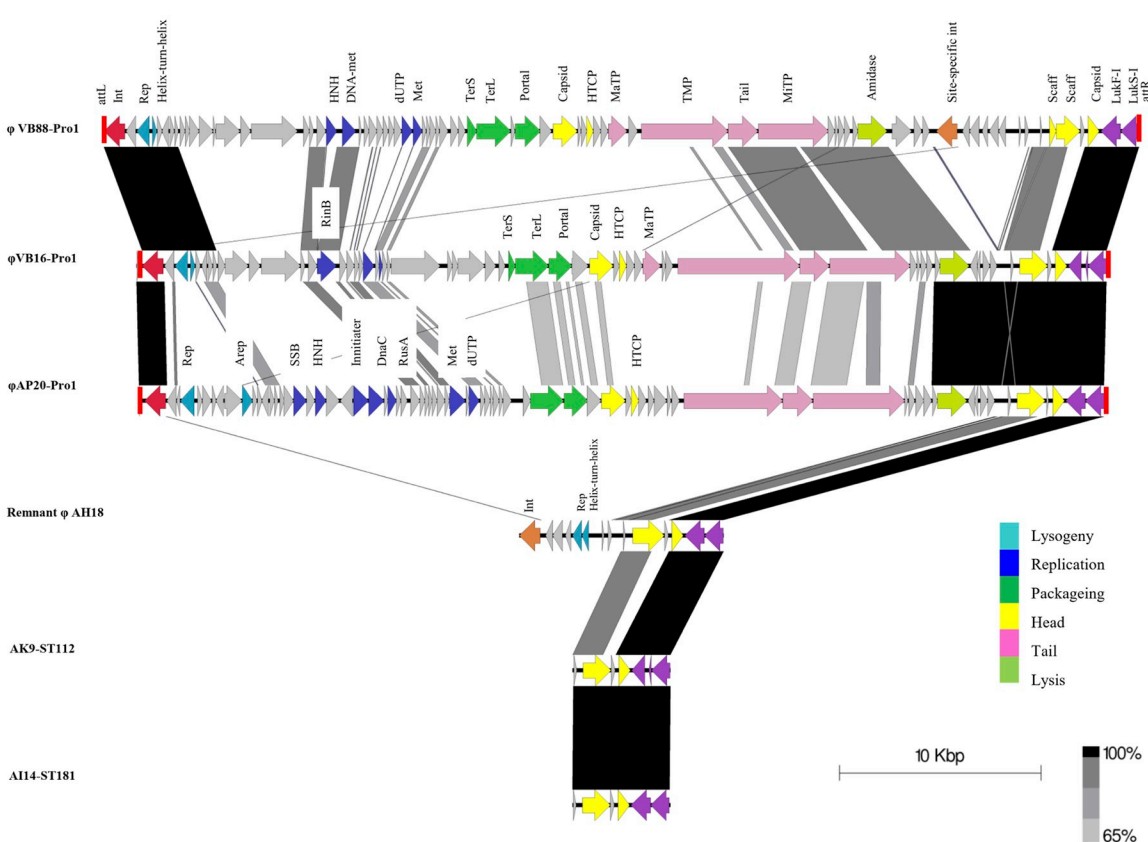

**Fig 3. Comparison of prophages φVB88-Pro1 (accession no. CP030715 position 1,158–53,190), φVB16-Pro1 (accession no. CP0301605 position 1,294,492–1,342,940), and φAP20-Pro1 (accession no. CP031561 position 764,687–812,888) of human isolates to remnant of φAH18-Pro1 (accession no. CP030374 position 2,192,044–2,202,245) and partial similar region found in strain AK9-ST112 (accession no. CP031603 position 2,153,065–2,157,877) and AI14-ST181 (accession no. CP031604 position 2,712,104–2,716,953).** *att*, putative attachment site left (L) and right (R); Int, integrase; Helix-turn-helix, XRE family protein; Rep, phage repressor; Arep, anti-repressor; HNH, HNH endonuclease; SSB, single-stranded DNA-binding protein; DNA met, DNA-cytosine methyltransferase; dUTP, dUTPase; Met, methyltransferase; RinB, transcriptional regulator RinB; Initiator, phage replication initiation protein; DnaC, DNA replication protein DnaC; RusA, Holliday junction resolvase; TerS, terminase small subunit; TerL, terminase large subunit; HTCP, head-tail adaptor protein; MaTP, major tail protein; TMP, phage tail tape measure protein; Tail, phage tail protein; MiTP, minor tail protein; Scaff, scaffold protein. Grey color arrows represent hypothetical protein and protein of unknown function. Figure was generated using Easyfig 2.1.

The two phages φVB88-Pro2 and φVB16-Pro2 from the human isolates were integrated between the *sufB* gene of the Fe-S cluster and a protease gene, and downstream of the haemo-lysin gene *hly*. The integrase of φVB88-Pro2 and φVB16-Pro2 shared 98–99% amino acid similarity to that of bacteriophage spT152 from *S. pseudintermedius* E139 (accession no. KX827369) (S5 Fig). Similarities were also observed for the head, tail and lysis modules among the prophages. The third type of prophage found in the human MRSP isolates, φVB16-Pro3, carried a recombinase of the resolvase/invertase family at the 5' end followed by complete phage structures. The recombinase of φVB16-Pro3 shared 92% amino acid similarity with an enzyme of *S. pseudintermedius* NA45-ST84 (accession no. CP016072), which also contained a prophage sequence similar to φVB16-Pro3 in its chromosome (S5 Fig) and with a similar integration site. The related prophage of NA45-ST84 and φVB16-Pro3 were inserted into a C4-dicarboxylate ABC transporter protein gene. Both prophages did not contain known viru-lence genes.

For the dog MRSP strains, φAK9-Pro1 and φAI14-Pro1 were identified, where φAK9-Pro1 was distinct from the other prophages because of its large size and structural organization (S5 Fig). The integrase of φAK9-Pro1 showed 100% amino acid identity to another integrase of *S. pseudintermedius* NA45. Although NA45 contained a similar prophage to φAK9-Pro1, the chromosomal integration site differed. The prophage of NA45 was integrated between a gene for ribosome-binding factor A and translation initiation factor 2 (*infB*), while φAK9-Pro1 was found between the genes for a hypothetical protein and *infB*. The second prophage in the dog isolates AI-14 (φAI14-Pro1) showed similarity to a prophage region of the canine *S. pseudintermedius* strains HKU10-03-ST308 (accession no. CP002439; S5 Fig). Both prophages were integrated into the late competence protein ComGA gene (*comGA*) and had 99.8% nucleotide similarity to each other, but the integrase gene of prophage in HKU10-03 had an overlapping gene annotation (S5 Fig). Putative virulence factors were not identified in either φAK9-Pro1 or φAI14-Pro1.

## Genomic structure of *S. pseudintermedius* PIs (SpPIs)

Island viewer and PHASTER predicted some overlapping region of putative PIs in the genomes of the ST181 isolates. Manual inspection of the sequences revealed a novel composite genomic island in AI14-ST181 and AP20-ST181, named SpPI-181 (Fig 4). This consisted of a 20-kbp region, which contained a 14 kbp PI and an incomplete 6 kbp genomic island that carried only some features of a PI. Flanking DRs predicted from PHASTER enclosed either the complete SpPI-181 structure (DR1) or the 14 kbp PI (DR2) (Fig 4). Chromosomal integration of SpPI-181 was situated between the guanosine monophosphate synthase gene (GMPS; *guaA*) and a hypothetical protein gene, a locus that was reported to contain a PI in other *Staphylococcus* spp. [32, 33]. The left integrase showed the highest amino acid similarity (76%) with an

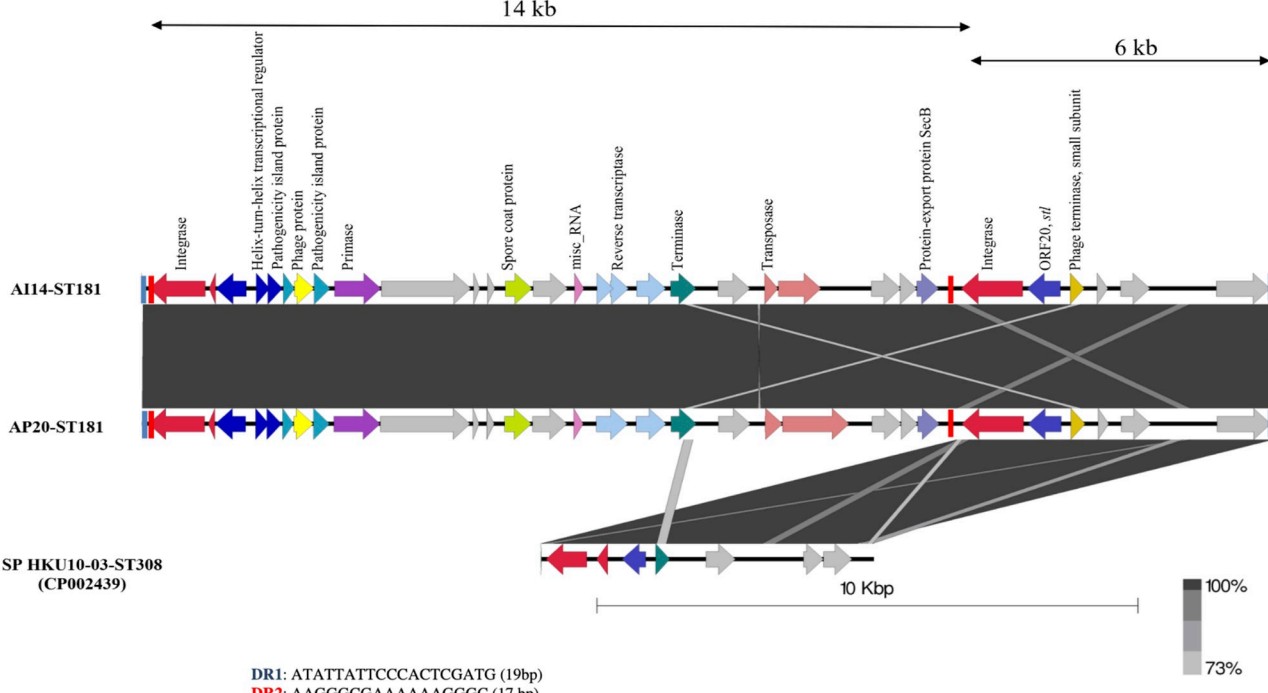

**Fig 4. Structure of SpPI-181 in AI14-ST181 and AP20-ST181 compared with the sequence of *S. pseudintermedius* HKU10-03-ST308.** Hypothetical proteins and proteins with unknown function are represented in grey color. Figure was generated using Easyfig 2.1.

integrase of *S. haemolyticus* (accession no. WP_037559001), while the right integrase showed 64% amino acid identity with an integrase of *S. pseudintermedius* HKU10-03-ST308. The 6 kbp island was also present in the *S. pseudintermedius* HKU10-03-ST308 genome at the same location (Fig 4). The 14 kbp island of SpPI-181 contained features of a PI, including integrase, helix-turn-helix transcriptional regulator, pathogenicity island protein, phage protein, primase, and a terminase small subunit gene. In addition to the *int* gene, the 6 kbp island of SpPI-181 contained a homolog of a *stl* repressor gene (ORF20) and a phage terminase small subunit gene. The accessory genes of SpPI-181 were the protein-export protein SecB (*secB*) and spore coat protein (*cot*), with a 88% amino acid similarity to the hypothetical protein in *S. simulans* (accession no. CP014016.2). No known virulence gene was identified in SpPI-181.

## Integrase genes detection in additional MRSP

The dissemination of the putative prophages and PIs detected in the sequenced MRSP strains was determined in additional *S. pseudintermedius* isolates from dogs (n = 61) and humans (n = 16) by PCR assay. To do so, 8 primer pairs specific to *int* genes of prophage and phage-associated elements were designed (S1 Table). For SpPI-181, two PCRs were used to detect the 14 kbp and 6 kbp PIs (Fig 4). The *int* of the remnant φAH18-Pro1 and that found in the accessory region of φVB88-Pro1 are identical and detected with the same primer pair (Fig 3). The majority (71/77) of the MRSP isolates contained one or two *int* genes, with six strains that did not contain any *int* (Table 2). The isolates belonging to clonal complex (CC) 45 had a greater diversity of phage types, and were represented by the presence of *int-2*, the combination of *int-1* and *int-2*, *int-3*, and *int-5*. The *int-2* without the presence of *int-1* representing the remnant φAH18-Pro1 was widely deposited in many STs (n = 28) without host specificity. φAK9-Pro1 (*int-6*) was specific to the dog MRSP strains belonging to ST112 and ST111. The isolates of ST183 from dogs and humans did not contain any of the investigated integrases. For the integrase detection of SpPI-181, the combination of *int-7* and *int-8* was specific to the isolates belonging to ST181. The remnant of SpPI-181 (*int-8*) was observed in 4 different STs of 3

**Table 2. Integrase of prophages and SpPI detected in 77 MRSP isolates belonging to different sequence types (ST)[a].**

| integrase | Phage | Dog (n = 61) | Human (n = 16) | Total[c] |
|---|---|---|---|---|
| *int-1* | φ VB16-Pro1 (ST112)/ φ VB88-Pro1 (ST45)/ φ AP20-Pro1 (ST181) | 3: ST181 = 2, ST112 = 1 | 1: ST112 = 1 | 4 |
| *int-2* | remnant φ AH18 -Pro1 (ST45) | 23: **ST45 = 14, ST116 = 1** [b], ST55 = 1, ST114 = 1, ST185 = 2, ST182 = 3, ST178 = 1 | 5: ST45 = 3, ST110 = 1, ST178 = 1 | 28 |
| *int-1* and *int-2* | φ VB88-Pro1 (ST45) | 18: **ST45 = 18** | 3: **ST45 = 2, ST113 = 1**[b] | 21 |
| *int-3* | φ VB88-Pro2 (ST45)/ φ VB16-Pro-2 (ST112) | 3: **ST45 = 3** | 1: ST169 = 1 | 4 |
| *int-4* | φ VB16-Pro3 (ST112) | 0 | 0 | 0 |
| *int-5* | φ AI14-Pro (ST181) | 1: **ST45 = 1** | 1: **ST113 = 1** | 2 |
| *int-6* | φ AK9-Pro1 (ST112) | 8: ST112 = 7, ST111 = 1 | 0 | 8 |
| *int-7* and *int-8* | SpPI-181 in AI14 and AP20 (ST181) | 4: ST181 = 4 | 3: ST181 = 3 | 7 |
| *int-8* | Remnant of SpPI (ST181) | 3: ST185 = 2, ST121 = 1 | 2: ST181 = 1, ST169 = 1 | 5 |
| No *int* | - | 4: ST183 = 2, ST169 = 1, ST125 = 1 | 2: ST183 = 2 | 6 |

[a] Six representative strains (AH18-ST45, VB88-ST45, AK9-ST112, VB16-ST122, AI14-ST181, and AP20-ST181) were not included.

[b] Clonal complex (CC) 45 including ST45, ST113 and ST116; Lineage from eBURST analysis are indicated in bold letter.

[c] Some isolates contained ≥ 2 integrase genes.

canine isolates belonging to ST185 (n = 2) and ST121 (n = 1), and a human MRSP strains belonging to ST181 and ST169.

## Discussion

We compared whole genome sequences of three pairs of MRSP strains from dogs and humans belonging to three major clones distributed in dogs and humans in Thailand. Genome variation of the same ST from different hosts was speculated by different PFGE patterns [34].

Genetic diversity was confirmed by pan and core genome comparative analysis, including genomic similarity, SNPs, MGEs, and surface protein genes. Isolates belonging to ST45 had the highest difference among the three pairs of isolates, which may indicate a high adaptive ability, a feature that could explain the successful spread of ST45 isolates in Thailand. The differences were mainly associated with the presence of two additional prophages as well as truncation of some of the CWA protein genes in the human ST45 isolate. According to knowledge obtained from MRSA studies, only some lineages can adapt to colonize animals and humans [35].

Overall, eight different prophages were identified in the six MRSP analysed. Seven of them were classified as members of *Siphoviridae* family and one (φ AK9-Pro1) was classified in the *Myoviridae* family based on the characteristics and structures used for bacteriophage nomenclature [36–38]. Phages of the *Siphoviridae* family consist of temperate phages and are the major phage family in staphylococci, while phages of *Myoviridae* family consist of lytic and chronic phages which are important for phage therapy [38]. Accessory phage genes may provide some advantage to bacterial species enhancing virulence like the *lukF/S-I* genes [36]. Prophages carrying *lukF/S-I* were distributed in three different dominant clones. The inserted virulence factors on the phage lytic module were suggested to optimize and control the expression of the *lukF/S-I* genes in *S. aureus* [37].

The prevalence of prophages in the investigated *S. pseudintermedius* was similar to that of coagulase-negative staphylococci (CoNS) that commonly contain only few prophages and PI [39]. The number of prophages in *S. pseudintermedius* correlated with sequence types [40]. As observed in *S. aureus*, the absence of a prophage in one isolate of the lineage would indicate prophage loss, while the presence of the same prophage in different strains suggests possible prophage transfer [41]. However, horizontal gene transfer (HGT) and modules exchanging of prophages in different clones only occur rarely and with low frequency [41]. The φVB88-Pro1, remnant of AH18-Pro1, and φVB88-Pro2 were present or absent in both dog and human MRSP strains belonging to CC45, indicating a frequent loss or acquirement of these elements. A host-specific distribution was observed for φAK9-Pro1, being present in all canine ST112 isolates but absent in VB16-ST112 from human. This result might indicate that φAK9-Pro1 was lost when ST112 isolates colonized human hosts. On the other hand, prophages of the φVB16-Pro2/φVB88-Pro2 group seem to occur more frequently in human MRSP isolates belonging to different STs (ST45, ST112, and ST169) than in dog MRSP isolates (ST45), indicating that isolates containing φVB16-Pro2/φVB88-Pro2 might have an advantage in colonizing humans. The loss and gain of prophages might contribute to the flexibility of bacteria to adapt in new niches and environments [41, 42]. However, the detection of a similar *int* sequence alone does not provide information about its location in the genome and other modules that can be different, as seen for the phage belonging to the φVB88-pro1/VB16-Pro1/φAP20-Pro1 group. In addition, it is expected that there will be other prophages with different *int* genes circulating among the MRSP isolates that were not present in our six reference genomes. More closed genomes from long-read sequencing technology are needed to elucidate the diversity of prophages in the *S. pseudintermedius* population.

Both integrases of the novel composite SpPI-181 were identified in all of the dog MRSP strains belonging to ST181, while only one integrase, indication for remnant of SpPI-181, was found in a human MRSP strains of ST181 and ST169. These results indicate that the composite element might not be stable and different SpPI-181 structures might be associated with dog and human colonizing. The integrase of pathogenicity islands in bacterial genomes are important for bacterial evolution and are associated with HGT [37]. Further study of the evolutionary, HGT, and host specificity of SpPI-181 should be performed.

A total of 18 CWA protein genes (*sps*) were first identified in *S. pseudintermedius* ED99 [31]. Variation in *sps* genes was observed in isolates of ST45, where the *spa* genes, *spsP* and *spsQ*, were found in the genome of AH18-ST45 but not in VB88-ST45. A previous study revealed that ST45 consisted of two distinct sublineages of *spa*-positive and *spa*-negative [43]. The gene content in the *spa* locus (*spsP* and *spsQ*) and flanking region was highly variable in the same and different lineage. Therefore, the *spa* locus was suggested as one of the hot spots region for recombination and genetic exchange in *S. pseudintermedius* [43]. The association of *spa* with host adaptation was shown in MRSA ST5 and ST151, where Spa proteins were lost in clones adapted to live in poultry and cattle [35]. In addition, the diversity of CWA protein genes, *spsJ*, was only observed in isolates from humans. Base insertion and deletion leading to a premature stop codon were found at the end of conserved regions near sequence variation regions (SDSD and STS) that tend to mutate under diversification selection [44]. These frameshift mutations truncated the gene and may affect its function. However, the function of each part of *spsJ* is still unknown [44].

In addition, deletion of the repeat region was observed in the *spsO* of VB88-ST45 isolates from human. Although *spsO* is a homolog of the serine-aspartate repeat protein C (*sdrC*) in *S. aureus*, its function is still unknown [31, 45]. This region in SpsO was considered as B repeats according to its structure [46]. The presence of a B repeat deletion was reported in the Sdr gene of *S. aureus* and was suggested to be the result of high rate mutation through slipped strand mispairing [46]. Bacterial tandem repeats are mostly located on flexible genes, such as surface proteins, and their hyper mutations are often deleterious but can also be responsible for adaptation to a new environment [46, 47]. Taken together, the diversity of CWA protein genes might reflect adaptation to dogs and humans since CWA proteins are supposed to interact with the host tissue. Similarly, variation in the surface proteins was observed previously in the livestock-associated *S. aureus* ST398 lineage adapted to humans, including gene truncation, nucleotide insertion and deletion [8].

The difference in the gene encoding transporter system and metabolic enzymes are suggested to be associated with bacterial survival in different hosts and environments [48]. Three specific SCC*mec* acquisition including ψSCC*mec*$_{57395}$ of ST45, SCC*mec*$_{AI16}$-SCC*czr*$_{AI16}$-CI of ST112 and SCC*mec*V(T) of ST181 are associated with their divergent evolution and support clonal expansion of the lineages [49]. Acquisition of *Tn*5405-like elements contributing multidrug resistance properties is identified as a crucial step of evolution in successful MRSP clones [29]. The restriction systems, including RM system, DpnII, and restriction endonuclease, are a barrier for HGT [42]. Nevertheless, the role of the CRISPR system in preventing HGT in *S. pseudintermedius* is not well elucidated [29]. The diversity of restriction systems in isolates of the same and different lineages of *S. pseudintermedius* has been reported previously [29, 40]. This diversity may be caused by the selection process leading to total gene loss of the RM system and might be transferred by MGEs, as they are mostly located on SCC*mec* elements [50]. The presence of different prophages and other MGEs in each lineage can be the result of these different RM systems.

It should be noted that the low number of isolates used in this study may limit the host-specific findings, while the time of colonization in the human nasal cavity was not known. A

large-scale comparative analysis of dog and human MRSP genomes may provide more insight and additional data. In addition, *S. pseudintermedius* isolates from human infections may be a good representative for the study of human adaptation. However, cases of human infections have not been reported in Thailand yet.

In conclusion, this study found several prophages that were distributed differently in MRSP isolates from dogs and humans, indicating active phage transfer within isolates from the same and different clones. Prophages were associated with virulence factors, genetic exchange, host specificity, and bacterial evolution. In addition, the diversity of CWA protein genes showed an association with different host colonization. The high mutation rate in the core genome, especially in ST45, a major clone, could also represent the ability to adapt in each environment. This finding expands the knowledge of the molecular adaptation of MRSP that is affected by different host colonization.

## Supporting information

**S1 Table. Primers used in this study.**
(PDF)

**S2 Table. Unique genes of human isolates predicted from Roary.**
(PDF)

**S3 Table. Information of prophages found in this study.**
(PDF)

**S1 Fig. Genomic similarity (%) of 6 representative strains derived from ANIb defaults in Jspecies.**
(TIFF)

**S2 Fig. Phylogenetic tree of SNPs found in all analysed genomes.** SNPs tree was constructed by CSIPhygeny 1.4.
(TIFF)

**S3 Fig. SNPs count compared in all isolates.** SNPs numbers were calculated by CSIPhygeny 1.4. Percentage of reference genome covered by all isolates: 81.9109943221414. Size of reference genome *S. pseudintermedius* NA45 was 2,841,212 bp.
(TIFF)

**S4 Fig. Partial alignment sequences of *spsO* of AH18 and VB88- ST45 presenting 78-bp deletion in tandem repeat regions.** Pairwise alignment was performed in Genious v 10.1.3 using Clustal W defaults setting.
(TIFF)

**S5 Fig.** Comparison of phage modules of φ VB88-Pro2, φ VB16-Pro2, and φ spT152 (A), φ VB16-Pro3 and sequence of *S. pseudintermedius* NA45 (B), φ AK9-Pro1 and sequence of *S. pseudintermedius* NA45 (C), and φ AI14-Pro1 with sequence of *S. pseudintermedius* HKU10-03- ST308 (D). att, putative attachment site left (L) and right (R); Int, integrase; Helix-turn-helix, XRE family protein; CI-like; CI-like protein; Rep, phage repressor; Arep, anti-repressor; Xis, exisionase; HNH, HNH endonuclease; Gam-like, bacteriophage Mu Gam-like protein; SSB, single-stranded DNA- binding protein; DNA-met, DNA-cytosine methyltransferase; RecT, recombinational prophage- associated DNA repair protein RecT; dUTP, dUTPase; Met, methyltransferase; RinB, transcriptional regulator RinB; Initiator, phage replication initiation protein; DnaB, DNA replication protein DnaB; DnaC, DNA replication protein DnaC; DnaD, DNA replication protein DnaD; XerC,Tyrosine recombinase XerC; RusA, Holliday junction

resolvase; TerS, terminase small subunit; TerL, terminase large subunit; HTCP, head-tail adaptor protein; MaTP, major tail protein; TMP, phage tail tape measure protein; Tail, phage tail protein; MiTP, minor tail protein; Scaff, scaffold protein. Figure was generated by Easyfig 2.1. (TIFF)

**S1 Data.**
(XLSX)

## Author Contributions

**Conceptualization:** Vincent Perreten, Nuvee Prapasarakul.

**Data curation:** Nathita Phumthanakorn, Sybille Schwendener, Valentina Donà.

**Funding acquisition:** Nuvee Prapasarakul.

**Investigation:** Nathita Phumthanakorn, Valentina Donà, Pattrarat Chanchaithong, Nuvee Prapasarakul.

**Methodology:** Nathita Phumthanakorn, Vincent Perreten, Nuvee Prapasarakul.

**Project administration:** Vincent Perreten.

**Resources:** Pattrarat Chanchaithong.

**Software:** Sybille Schwendener, Valentina Donà.

**Supervision:** Vincent Perreten.

**Validation:** Nathita Phumthanakorn, Sybille Schwendener, Valentina Donà, Vincent Perreten, Nuvee Prapasarakul.

**Visualization:** Nathita Phumthanakorn, Sybille Schwendener, Valentina Donà.

**Writing – original draft:** Nathita Phumthanakorn, Sybille Schwendener.

**Writing – review & editing:** Vincent Perreten, Nuvee Prapasarakul.

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
