## [Decision Letter · Decision Letter 0]

3 Jun 2021

PONE-D-21-13891

Genomic insights into methicillin-resistant Staphylococcus pseudintermedius isolates from dogs and humans of the same sequence types reveals diversity in prophages and pathogenicity islands.

PLOS ONE

Dear Dr.Prapasarakul,

Thank you for submitting your manuscript to PLOS ONE. After careful consideration, we feel that it has merit but does not fully meet PLOS ONE’s publication criteria as it currently stands. Therefore, we invite you to submit a revised version of the manuscript that addresses the points raised during the review process.

Your manuscript has been reviewed by two experts and a minor revision is suggested. Please follow the reviewers' comments and make the necessary revision.

We look forward to receiving your revised manuscript.

Kind regards,

Yung-Fu Chang

Academic Editor

PLOS ONE

Journal Requirements:

2. In your Methods section, please provide additional details regarding participant consent from the owners of the animals. In the ethics statement in the Methods and online submission information, please ensure that you have specified (1) whether consent was informed and (2) what type you obtained (for instance, written or verbal). If the need for consent was waived by the ethics committee, please include this information.

Reviewers' comments:

Reviewer's Responses to Questions

**Comments to the Author**

1. Is the manuscript technically sound, and do the data support the conclusions?

Reviewer #1: Yes

Reviewer #2: Yes

2. Has the statistical analysis been performed appropriately and rigorously? 

Reviewer #1: N/A

Reviewer #2: Yes

3. Have the authors made all data underlying the findings in their manuscript fully available?

Reviewer #1: Yes

Reviewer #2: Yes

4. Is the manuscript presented in an intelligible fashion and written in standard English?

Reviewer #1: Yes

Reviewer #2: Yes

5. Review Comments to the Author

Reviewer #1: A thorough description of the MRSP genome from both dogs and humans. This manuscript is well written, informative and gives initial insight into an interesting gap in knowledge about host adaptation in this important pathogen. Thank you for this manuscript.

I only have minor comments which are typographical in nature.

Line 69 - important c.f. importance

Line 93 - Should this be 'stored' in TSB?

Line 144 - This sentence doesn't appear to be complete?

Line 146 - sp. detection

Line 177 - '...difference in...' c.f. '...different of...'

Line 200 - isolate should be singular

Line 279 - Should this be Figure 3 references for the prophage in these strains?

Line 371-2 - Table 2 appears to show that int-6 is present in both ST112 and ST111 but this sentence suggests it is just in ST112. There are also 4 ST183 isolates in the table with no integrases (n=2 canine, n=2 human).

Line 374-376 - int8 is listed in the table as being present in canine ST185 and ST121 (n=3 total) but this sentence appears to suggest that it is only present in human isolates.

Line 382 - sp. eBURST

Reviewer #2: The present study by Phumthanakorn et al reports the whole genome sequencing and comparative genomics of six methicillin-resistant Staphylococcus pseudintermedius (MRSP) isolates from dogs (n=3) and humans (n= 3), belonging to ST45, ST112, and ST181. The main objective of the study was to determine the genetic similarities and differences between human and animal MRSP strains of a same sequence type (ST). The authors have analyzed the integrated prophages, pathogenicity island, plasmids, transposable elements and various genes linked to antibiotic resistance and virulence in the sequenced strains. Similar to previous studies, some of the eight prophages identified in this study were ST-specific. All sequences have been submitted to the NCBI GenBank, and findings are nicely presented in figures and tables. Overall, the study is important and I do not have major comments. Here are my minor comments-

1. As mentioned in the paper “AH18-ST45 harbored two additional genes (spsP and spsQ) that were absent in VB88-ST45 and a different spsO genes”. This finding needs to be discussed in detail in the light of a recent study by Zukancic et al 2020 (mSphere) where spsP/spsQ gene deletion has been studied across all S. pseudintermedius sequence types.

2. Method section: Line 82-87. There are repetitive sentences about the isolation source of the dog and human strains sequenced in this study. It appears to be a typing error.

6. PLOS authors have the option to publish the peer review history of their article (what does this mean?). If published, this will include your full peer review and any attached files.

Reviewer #1: No

Reviewer #2: No

---

## [Author Response · Author response to Decision Letter 0]

8 Jun 2021

Dear editor:

 Herewith please find the responses to the comments of our manuscript entitled “Genomic insights into methicillin-resistant Staphylococcus pseudintermedius isolates from dogs and humans of the same sequence types reveals diversity in prophages and pathogenicity islands”. As you will see, we have responded to all the reviewers' recommendations and journal requirements, including journal format, references, and ethics statement, and modified the manuscript accordingly.

 We thank you the reviewers for your time and inputs. We trust that our additional data and revision are satisfactory for reconsidering the manuscript for publication in PLOS ONE journal. 

Sincerely yours,

Nuvee Prapasarakul

Reviewers' comments:

Review Comments to the Author

Reviewer #1: A thorough description of the MRSP genome from both dogs and humans. This manuscript is well written, informative and gives initial insight into an interesting gap in knowledge about host adaptation in this important pathogen. Thank you for this manuscript.

Response: Thank you very much for your valuable comments. We apologize for the misspelled words and incomplete sentences. We have corrected the manuscript following your advice.

I only have minor comments which are typographical in nature.

Line 69 - important c.f. importance

Response: importance was replaced with important

Line 93 - Should this be 'stored' in TSB?

Respopnse: "stored" has been addded

Line 144 - This sentence doesn't appear to be complete?

Response: The sentence has been rewritten as follows: " The presence of integrase genes of prophages- and PI-associated gene was determined for a previously collected isolates (n=77). 

Line 146 - sp. Detection

Response: detection has been corrected

Line 177 - '...difference in...' c.f. '...different of...'

Response: '...different of...' has been replaced with '...difference in...'

Line 200 - isolate should be singular

Response: it has been placed in singular.

Line 279 - Should this be Figure 3 references for the prophage in these strains?

Response: Fig 2 was replaced with Fig 3. 

Line 371-2 - Table 2 appears to show that int-6 is present in both ST112 and ST111 but this sentence suggests it is just in ST112. There are also 4 ST183 isolates in the table with no integrases (n=2 canine, n=2 human).

Response: The results of Table 2 were corrected and the manuscript was revised in line 372-374, and 375-377 accordingly. Line 374-376 “φAK9-Pro1 (int-6) was specific to the dog MRSP strains belonging to ST112 and ST111. The four isolates of ST183 from dogs and humans did not contain any of the investigated integrases.

Line 374-376 - int8 is listed in the table as being present in canine ST185 and ST121 (n=3 total) but this sentence appears to suggest that it is only present in human isolates.

Response: Line 377-379 “The remnant of SpPI-181 (int-8) was observed in 4 different STs of 3 canine isolates belonging to ST185 (n=2) and ST121 (n=1), and a human MRSP strains belonging to ST181 and ST169.”

Line 382 - sp. eBURST

Response: eBURST was corrected

Reviewer #2: The present study by Phumthanakorn et al reports the whole genome sequencing and comparative genomics of six methicillin-resistant Staphylococcus pseudintermedius (MRSP) isolates from dogs (n=3) and humans (n= 3), belonging to ST45, ST112, and ST181. The main objective of the study was to determine the genetic similarities and differences between human and animal MRSP strains of a same sequence type (ST). The authors have analyzed the integrated prophages, pathogenicity island, plasmids, transposable elements and various genes linked to antibiotic resistance and virulence in the sequenced strains. Similar to previous studies, some of the eight prophages identified in this study were ST-specific. All sequences have been submitted to the NCBI GenBank, and findings are nicely presented in figures and tables. Overall, the study is important and I do not have major comments. Here are my minor comments-

Response: Thank you very much for the comments and recommendation.

1. As mentioned in the paper “AH18-ST45 harbored two additional genes (spsP and spsQ) that were absent in VB88-ST45 and a different spsO genes”. This finding needs to be discussed in detail in the light of a recent study by Zukancic et al 2020 (mSphere) where spsP/spsQ gene deletion has been studied across all S. pseudintermedius sequence types.

Response: The following sentence has been included into the Discussion section Line 444-449: “A previous study revealed that ST45 consisted of two distinct sublineages of spa-positive and spa-negative [43]. The gene content in the spa locus (spsP and spsQ) and flanking region was highly variable in the same and different lineage. Therefore, the spa locus was suggested as one of the hot spots region for recombination and genetic exchange in S. pseudintermedius [43].” The Reference 43 is from Zukancic A, Khan MA, Gurmen SJ, Gliniecki QM, Moritz-Kinkade DL, Maddox CW, et al. Staphylococcal protein A (spa) locus is a hot spot for recombination and horizontal gene transfer in Staphylococcus pseudintermedius. mSphere. 2020;5(5):e00666-20. doi: 10.1128/mSphere.00666-20.

2. Method section: Line 82-87. There are repetitive sentences about the isolation source of the dog and human strains sequenced in this study. It appears to be a typing error.

Response: The sentence was corrected as follows Line 82-85: - “Canine MRSP strain AH18 (ST45) was isolated from nasal mucosa, and MRSP strain AK9 (ST112) and AI14 (ST181) were isolated from groins. Human MRSP strain VB16 (ST112) and VB88 (ST45) were recovered nares from small animal veterinarians, and MRSP AP20 (ST181) was isolated from nares of a dog owner.” 

Journal Requirements: 

 In your Methods section, please provide additional details regarding participant consent from the owners of the animals. In the ethics statement in the Methods and online submission information, please ensure that you have specified (1) whether consent was informed and (2) what type you obtained (for instance, written or verbal). If the need for consent was waived by the ethics committee, please include this information.

Response: All strains were obtained from our previous study. All sample collections from humans and animals were formerly approved by the ethics statement in the method section of the manuscript as follows: (Line 96-99) " The strains were obtained from a previous study [3], where the sampling of dogs were performed after the informed consent were completed and signed by the dog’ owners. The ethical statement of dogs and humans were approved as previously described [3]."

---

## [Decision Letter · Decision Letter 1]

25 Jun 2021

Genomic insights into methicillin-resistant Staphylococcus pseudintermedius isolates from dogs and humans of the same sequence types reveals diversity in prophages and pathogenicity islands.

PONE-D-21-13891R1

Dear Dr. Prapasarakul,

We’re pleased to inform you that your manuscript has been judged scientifically suitable for publication and will be formally accepted for publication once it meets all outstanding technical requirements.

Kind regards,

Yung-Fu Chang

Academic Editor

PLOS ONE

Additional Editor Comments (optional):

Reviewers' comments:

Reviewer's Responses to Questions

**Comments to the Author**

1. If the authors have adequately addressed your comments raised in a previous round of review and you feel that this manuscript is now acceptable for publication, you may indicate that here to bypass the “Comments to the Author” section, enter your conflict of interest statement in the “Confidential to Editor” section, and submit your "Accept" recommendation.

Reviewer #1: All comments have been addressed

Reviewer #2: All comments have been addressed

2. Is the manuscript technically sound, and do the data support the conclusions?

Reviewer #1: Yes

Reviewer #2: Yes

3. Has the statistical analysis been performed appropriately and rigorously? 

Reviewer #1: Yes

Reviewer #2: Yes

4. Have the authors made all data underlying the findings in their manuscript fully available?

Reviewer #1: Yes

Reviewer #2: Yes

5. Is the manuscript presented in an intelligible fashion and written in standard English?

Reviewer #1: Yes

Reviewer #2: Yes

6. Review Comments to the Author

Reviewer #1: (No Response)

Reviewer #2: (No Response)

7. PLOS authors have the option to publish the peer review history of their article (what does this mean?). If published, this will include your full peer review and any attached files.

Reviewer #1: No

Reviewer #2: No

---

## [Editor Report · Acceptance letter]

13 Jul 2021

PONE-D-21-13891R1 

Genomic insights into methicillin-resistant *Staphylococcus pseudintermedius* isolates from dogs and humans of the same sequence types reveals diversity in prophages and pathogenicity islands 

Dear Dr. Prapasarakul:

I'm pleased to inform you that your manuscript has been deemed suitable for publication in PLOS ONE. Congratulations! Your manuscript is now with our production department. 

Kind regards, 

on behalf of

Dr. Yung-Fu Chang 

Academic Editor

PLOS ONE